# The Future of Careers at the Intersection of Climate Change and Public Health: What Can Job Postings and an Employer Survey Tell Us?

**DOI:** 10.3390/ijerph17041310

**Published:** 2020-02-18

**Authors:** Heather Krasna, Katarzyna Czabanowska, Shan Jiang, Simran Khadka, Haruka Morita, Julie Kornfeld, Jeffrey Shaman

**Affiliations:** 1Columbia University Mailman School of Public Health, 722 W. 168th St., 1003, New York, NY 10032, USA; sj2921@cumc.columbia.edu (S.J.); sk4537@cumc.columbia.edu (S.K.); hm2487@cumc.columbia.edu (H.M.); jk3924@cumc.columbia.edu (J.K.); jls106@cumc.columbia.edu (J.S.); 2Care and Public Health Research Institute (CAPHRI), Maastricht University Department of International Health, Faculty of Health, Medicine and Life Sciences, Maastricht University, 6211 Maastricht, The Netherlands; kasia.czabanowska@maastrichtuniversity.nl; 3Institute of Public Health, Faculty of Health Sciences, Jagiellonian University, 31-007 Krakow, Poland; 4National Institute of Public Health, 00-791 Warsaw, Poland

**Keywords:** climate change, health workforce, workforce planning, competencies, public health education

## Abstract

Climate change is acknowledged to be a major risk to public health. Skills and competencies related to climate change are becoming a part of the curriculum at schools of public health and are now a competency required by schools in Europe and Australia. However, it is unclear whether graduates of public health programs focusing on climate change are in demand in the current job market. The authors analyzed current job postings, 16 years worth of job postings on a public health job board, and survey responses from prospective employers. The current job market appears small but there is evidence from job postings that it may be growing, and 91.7% of survey respondents believe the need for public health professionals with training in climate change may grow in the next 5–10 years. Current employers value skills/competencies such as the knowledge of climate mitigation/adaptation, climate-health justice, direct/indirect and downstream effects of climate on health, health impact assessment, risk assessment, pollution-health consequences and causes, Geographic Information System (GIS) mapping, communication/writing, finance/economics, policy analysis, systems thinking, and interdisciplinary understanding. Ensuring that competencies align with current and future needs is a key aspect of curriculum development. At the same time, we recognize that while we attempt to predict future workforce needs with historical data or surveys, the disruptive reality created by climate change cannot be modeled from prior trends, and we must therefore adopt new paradigms of education for the emerging future.

## 1. Introduction

Climate change is acknowledged to be a major threat to public health [1,2]. Just as public health practice must constantly adapt to emerging viral outbreaks, non-communicable diseases, or other health threats, it must also be prepared for the diverse threats to human health posed by climate change. Several reports and large-scale commissions [3,4,5,6,7,8,9,10] point to the need for training for the health workforce, including the public health workforce, in skills and content to help lead efforts to mitigate and manage the impacts of climate change on health.

A 2008 report by the Association of Schools and Programs of Public Health (ASPPH) mentioned climate change as a key, new area of public health education [11]. The 2016 Council on Education in Public Health (CEPH) competencies for public health education include areas of focus, which allow public health professionals to protect human health from climate change impacts, such as analyzing data, discussing structural bias, assessing “population needs, assets and capacities that affect communities’ health” and “applying systems thinking” [12].

Many competencies required for environmental health science students, such as “approaches for assessing, preventing and controlling environmental hazards that pose risks to human health and safety” [13] are applicable to climate change. However, knowledge of climate change specifically is not yet a core competency of public health degrees in the United States. New initiatives exist, such as the Global Consortium on Climate and Health Education, which now has 193 members [14] and recently proposed a set of Core Climate and Health Competencies for Health Professionals [15]. Additionally, the Association of Schools of Public Health in the European Region (ASPHER)’s 2018 Competencies does list climate change as a competency within “Population Health and Its Material-Physical, Radiological, Chemical and Biological-Environmental Determinants” [16] and the Council of Academic Public Health Institutions Australia (CAPHIA)’s Foundation Competencies for Public Health Graduates in Australia include “identify and describe the impacts of climate change and implications for ecologically sustainable development” and “climate change theory” [17].

To further identify existing research on the skills, competencies, and job market for individuals with training in both public health and climate change, we conducted a brief narrative review of the literature, primarily focusing on a keyword search of Google Scholar of “climate change” AND “public health” AND “workforce”, which yielded 28,100 results, and “climate change” AND “public health” AND “jobs”, which yielded 86,000 results; we also conducted a search of Pubmed.com for “public health education” AND (“climate change” OR “global warming”). Inclusion criteria included a focus on expected hiring needs for professionals with training in both climate change and public health. Articles that did not include information related to issues with workforce or training needs were excluded.

To identify competencies needed in a future workforce, and to ensure training aligns with labor market demand, it is accepted practice to rely on input from public health employers and organizations. Many ASPPH competencies are based on “blue ribbon panels” of employers [18], as are the Core Competencies for Public Health Professionals developed by the Council on Linkages Between Academia and Public Health Practice [19]. Similar employer input is needed to understand which skills current employers expect of public health graduates with respect to climate change. While employer surveys have been conducted in several public health workforce research articles [20,21,22,23,24,25], analysis of job postings-a potential key indicator of current employer requirements-has only rarely been used in the public health field [26,27,28]; this, combined with a survey of employers, can provide a fuller labor market analysis than has been conducted in the past.

Through our analyses, we can attempt to estimate current and future hiring trends for public health professionals with training in climate change-related competencies, as well as continue to identify the training needed to help address the threat of climate change. For those institutions creating new training programs focusing on both climate change and public health, it will be important to assess whether their graduates will be in demand in the labor market, and if so, which sectors are most interested in hiring candidates with these skills. We attempt to address the questions: Which employers currently seek graduates with training in both climate change and public health; and is the demand for such graduates likely to grow?

## 2. Materials and Methods

In order to best discern whether there is a growing need for professionals with a combination of training in both public health and climate change, the researchers conducted an analysis of current job postings; and to create projections into the future, we conducted an analysis of 16 years worth of job postings in a public health job board. Finally, we conducted a survey of potential employers of public health graduates focusing on climate change to ask for their projections of the skills needed for this future workforce.

### 2.1. Data Sources

#### 2.1.1. Analysis of Current Job Postings on Indeed.com (Job Board Aggregator)

In order to determine what types of organizations are currently hiring candidates in the USA with a combination of skills or experience in both public health and climate change, on 14 December 2019, the authors conducted a search of Indeed.com, a job board aggregator, which “crawls” multiple job posting websites to gather millions of job postings into one, searchable database [29]. The rationale for searching Indeed.com is that job postings on the site are pulled from a broad range of thousands of job posting sites (including organizations’ job sites as well as job boards), providing a snapshot of any jobs—not only within traditional public health organizations—that include a combination of relevant keywords, allowing an assessment of the scope of the existing job market and whether current jobs fit the training of public health graduates. Indeed.com allows for Boolean search operators. The authors searched for jobs with the following keywords: (“climate change” OR “global warming”) AND (“public health” OR “environmental health” OR epidemiology OR “health policy”). A total of 172 jobs were found on Indeed.com; duplicates were removed, for a total of 159 positions. We then conducted a “scrape” (download) of the results using a commercially available web scraping tool called Scrapestorm [30], to identify the industries/sectors of the jobs with this combination of phrases. The Indeed.com main site primarily identifies jobs in the United States.

The resulting Excel file of organizations, job titles and descriptions, were then analyzed using the National Cancer Institute’s SOCcer (Standardized Occupation Coding for Computer-assisted Epidemiological Research) system [31], “a publicly available application that was developed to assist epidemiological researchers incorporate occupational risk into their studies”, to create Standard Occupational Classification [32] codes for the downloaded search results; those results with a lower degree of certainty in the automated coding system were hand-coded by the authors.

The industries/sectors of the employer organizations were also hand-coded, using a taxonomy in alignment with the new ASPPH employment outcomes data collection [33]. For context, an Indeed.com search of only the keywords “climate change” OR “global warming” conducted on December 19, 2019, found 2423 results. Thus, approximately 6.6% of the search results on Indeed.com related to climate change have an overlap with public health (159 of 2423). An Indeed.com search for (“public health” OR “environmental health” OR epidemiology OR “health policy”) on 27 December 2019, found 37,490 jobs, so approximately 0.4% of public health-related jobs also mentioned climate change or global warming.

#### 2.1.2. Analysis of 16 Years Worth of Job Postings on Publichealthjobs.org

The authors were provided access to 32,093 job postings posted into the free job board managed by ASPPH, publichealthjobs.org (previously publichealthjobs.net) dating from 17 July 2003–23 April 2019 [34]. This job board is frequently used by public health employers; it receives approximately 8.16% of all Internet traffic for the search terms “public health jobs” [35] and has been used for other analysis [26]. Of the 30,991 job postings for which the geographic location was known, 11.2% were from countries outside the United States. Unlike Indeed.com, which searches for job postings across numerous job posting websites throughout the Internet, the Publichealthjobs.org website requires employers to manually post their positions into the site, creating a self-selecting group of job postings that are specific to public health. The job description and requirements sections of the job postings were searched for the keywords “climate change” OR “global warming”. Duplicates were removed. An analysis of the proportion of all postings that included either of the target phrases was conducted on a year over year basis from 2003 to 2019, using R coding [36].

### 2.2. Survey of Relevant Employers

In order to assess the views of current employers who are likely to need candidates with training in both public health and climate change, the authors created an online survey using Qualtrics [37]. The survey questions were created through consultation with experts in both climate change and public health education, and included both closed-ended and open-ended questions (see Appendix A for survey questions). Questions regarding specific competencies were based on the current curriculum of Columbia University’s Climate and Health Certificate program. The survey and outreach methods were approved by the Columbia Human Subjects Review Board. Respondents were identified by the Columbia University Mailman School of Public Health Office of Career Services, which utilized its existing job posting database, a directory of approximately 5900 contactable employers who had posted a job or internship with Columbia University School of Public Health, or otherwise engaged with the career center, since 2012. These records are maintained using a secure vendor software hosted by the GradLeaders [38] company, and are accrued in a variety of ways: career services staff members conduct ongoing, targeted outreach by attending conferences and events such as the American Public Health Association conference, career fairs (including those focused on environment and sustainability), professional association memberships, online directories, leveraging faculty connections, and connecting to recruiters and alumni via LinkedIn.com and other social media platforms. Staff focused employer outreach efforts using input from ongoing surveys of students and engagement with academic departments and student organizations. A subset of 450 employer contacts from the jobs database was identified based on past job postings with keywords such as “climate change”, as well as by targeting employers in industries and sectors related to environmental health.

Additional, new contacts were identified by using specific keyword searches on LinkedIn such as: Job Title search for (sustainability OR resilience OR mitigation OR adaptation OR carbon) and the general keywords of “Climate Change” AND “health”; and attempts were made to diversify industries of respondents. This allowed the authors to identify 100 new contacts; of these contacts, 12 were directly contacted via LinkedIn “InMail” messages and 51 by using publicly available information; 37 could not be contacted directly. Twenty-one alumni of the Columbia School of Public Health’s Climate Change and Health Certificate program were also surveyed. Three contacts were referred by faculty at Columbia. A total of 537 active contacts were identified from all sources; contacts were primarily based in the USA.

The survey was distributed in January, 2019, with two reminders sent, once in January and once in March, 2019, and the survey was closed on 9 April 2019. Survey respondents were offered an opportunity to win a $50 gift card as an incentive for responding to the survey, and they were also encouraged to forward the survey to others in their network. Ninety seven individuals responded. Ten respondents were excluded because they were current graduate students or postdoctoral researchers, as opposed to professionals employed in the field. In addition, the survey was forwarded to other contacts in many cases, and a link to the survey was also posted on several online discussion boards including the Planetary Health Online Community and Planetary Health Education Subgroup on Hylo [39]. Contrasting the survey recipients with responders, we found that 75 respondents came from our survey outreach and 12 were not on the survey distribution list. Of the 87 respondents, five were US-based international non-governmental organizations, one was a multilateral government organization, one was an international consulting firm, one was a US government agency focusing on global health, and seven were NGOs and corporations based in other countries including China, Mexico, the UK, Kenya, Haiti, and Ecuador. Thus 15 of 87, or 17% of the respondents were international.

A statistical analysis of the responses was conducted. To evaluate the perceived usefulness of skills among employers in the public health field, we designed a mixed version of questions in which the responses are ordinal consisting of seven levels or text. The survey questionnaire comprises fourteen Likert-scale items to assess the usefulness of specific skills; in the later analysis stage we removed the “other” category, so only thirteen were left for the factor construct. We used qualitative methods to analyze the information from the open-ended responses. For the ordinal Likert-scale data, we first measured the internal consistency of the questionnaire, which was performed using the whole sample with Cronbach’s α values reported to be ≥0.60. Then we conducted a frequency description to identify if there was any ceiling effect or floor effect in the data. Finally, we used exploratory factor analysis and confirmatory factor analysis to identify the internal structure of the inventory. The factor analysis [40] is made up of two fundamental stages: (1) estimating the number of factors that should be extracted to represent the variability of the skillsets efficiently and (2) interpreting the meaning of the extracted factors and representing them in terms of theoretical structures that reflect the skillsets dimensions/sub-domains. In the analysis, factor loadings above ±0.40 were retained and listed in Table 4. We also assessed the trend of the annual number of public health job postings mentioning climate change or global warming as a function of year using Poisson regression. The total annual number of jobs was specified as an offset, and cross validation using continuous subsets of the total record was performed to determine if the results are unduly sensitive to a specific year or years. The descriptive and inferential statistical analyses were conducted using SPSS 24.0 [41] and R [36].

## 3. Results

### 3.1. Literature Review

Overall, there are many articles on the intersection between “climate change” and “public health,” but relatively little on labor market projections. Several articles directly mentioned how public health nurses or health professionals can become involved in climate change response, prevention, adaptation, and mitigation, policy [42], risk management, disaster preparedness, vector-borne diseases, heat-related diseases, the evidence base for climate change adaptation, etc. [8,9,43,44]. One article focused on elements of workforce development including “undergraduate through postgraduate training” in health, professional development of existing workforce, and training of policy-makers [44]. There were three articles on the Australian response to climate change events such as bushfires, extreme heat, and poor air quality, as well as rural health services [5,7,45]. Other articles mention the training needs of governmental public health workers relating to climate change [46], or specific sub-areas of training such as nutrition [47], or the importance of communication [48], or focus generally on why climate change training is needed in public health education [49].

Several articles provided action plans related to climate change and public health, which would require workforce training [50,51]. These examples include diagnosing and investigating health problems and hazards; monitoring health status to identify and solve community health problems; focusing on disaster preparedness [4]; dealing with emerging infectious diseases influenced by climate change [4]; informing, educating, and empowering the public on these issues; evaluating the intervention effectiveness of population-based health services; and monitoring workforce strain due to climate change [52]. Overall, it is difficult to find quantitative public health employment data, but many of the articles mention the importance of training, workforce development, and education to prepare and integrate climate change into public health efforts.

#### 3.1.1. Analysis of Current Job Postings on Indeed.com

The search of job postings from Indeed.com yielded the following distribution by industry:

corporation 32 (20%); nonprofit 76 (47.8%); government 17 (10.7%); and university/academia 34 (21.4%). In terms of occupational codes, the occupations with the largest numbers represented in the data set are listed in Table 1.

It is worth noting that the Standard Occupational Classifications do not include “community organizer”, “grassroots activist”, or “campaign organizer” as categories, so positions with these titles—the largest single group of positions in the data set—were coded as “Community and Social Service Specialists, All Other”. There were a total of 17 faculty positions, 12 within schools of public health, and 5 in environmental or biological sciences. Environmental and occupational health roles—those most likely to be a fit for graduates with a Master’s degree in public or environmental health—totaled 14 positions out of 159. Other common occupations included attorneys (primarily at government agencies related to environmental protection as well as legal advocacy nonprofits), public relations and fundraising, sales, and engineering roles. These data suggest that pursuing doctoral-level education, or combining a public health degree with either law or engineering, might best qualify candidates with an interest in both public health and climate change in today’s job market, at least in the USA.

#### 3.1.2. Analysis of 16 Years Worth of Job Postings on Publichealthjobs.org

The proportion of the 32,093 jobs from publichealthjobs.org from July 2003–April 2019, which mention either “climate change” or “global warming” consistently was a very small percentage of the total, but the percentage increased over this time period (*p* < 0.0001, Poisson regression). Cross validation found this trend to be positive and statistically significant for all 12-year or longer continuous subset time periods. The data can be seen in Table 2 and is illustrated in Figure 1.

We can observe that a salient change occurred over time on jobs related to climate change from Table 2. Overall, the total number of jobs increased since 2006, and the variability remained stable since then.

#### 3.1.3. Survey of Relevant Employers

As is often the case with surveys, the survey responders did not fully reflect the recipient population. In particular, government agencies and universities responded at a higher rate than the survey recipient population, while corporations, hospitals, and nonprofits responded at a lower rate (see Table 3). Comparing the survey recipients and respondents with those organizations that were actively posting positions in Indeed.com related to both climate change and public health, we can see that the populations were not quite the same; the Indeed.com search found a comparable percentage of corporations, a higher percentage of universities and nonprofits, and a lower percentage of government agency positions in comparison with the survey recipients and responses. Therefore, it is difficult to determine whether the survey is an accurate representation of the organizations currently hiring public health graduates.

With this limitation in mind, we might still gather some conclusions. Fifty of Seventy three (68.5%) of the responders who answered the question, “Has your organization hired people with a Master of Public Health or PhD in Public Health in the past” responded “yes”. Eighty six individuals responded to the question, “Do you expect the need to hire people with a background in climate and public health to grow in your organization in the next 5–10 years?” and of these, 33 indicated “yes”, 34 “maybe”, 6 “no”, and 13 “don’t know”. Excluding the “don’t know” responders, we could determine that 91.7% of respondents believed that there might be a need for public health and climate change-trained individuals in their organizations in the future.

In addition, an analysis of the thirteen-item Likert scale questions regarding skills, which would be useful to the employer organization, was conducted. See the frequency of responses in Figure 2.

To standardize the questionnaire data, we considered the numeric data and text data separately. For numeric data, we found that the 13-item Likert scale response shows a high internal consistency of 0.879, which is described by Cronbach’s alpha. From Figure 2, it is shown that no ceiling effect or significant floor effect was detected, suggesting it should be well-qualified as a valid measure of skill outcomes for public health employers.

A three-factor solution (all loadings ≥0.40) showed the best model fit to the survey data set. The Scree plot of the final exploratory factor analysis (EFA) solution is shown in Figure 3; we can observe that the eigenvalues of the model dropped below 1.0 when the component number reached 4, which is acknowledged as the rule of thumb cut-off point in deciding the internal structure. Thus, we set our final internal structure as a 3-factor EFA solution; this solution explained 70.16% variance by these three extracted factors and represents 13 items selected from the scale (only Likert Scale questions were included; text question and the “other” category question were filtered). In Table 4, all factor loadings were within the range of 0.456–0.928.

Only two items had a cross-loading on more than one item with loadings >0.50 (Item 9 and Item 10), and we followed guidelines and discarded them in the final model. As shown in Table 4, the proposed model structure includes three dimensions and 13 items.

After the psychometric validation, we finalized the model with a 3-factor structure, with 11 items, and labeled them according to the theoretical context of each question. The first category was labeled as Population Health Exposure, included six items covering a range from population health analytical skills to general understanding of research methods, and also had a strong consistency of 0.83 (Table 5). The second category, in particular, targeted at the Climate-Related Knowledge and its intercorrelation with health status, showed a high consistency of around 0.9. The final category separates two Statistical Programming Language skills from other concrete skills, and included the two most popular statistical programming tools, R and SAS, which also retained a Cronbach’s alpha value of 0.76.

Open-ended comments in response to the question “What expertise or skills do you think will be needed to address the issue of climate change and human health in the next 10–20 years?” were coded using qualitative analysis methods, using categories identified by two of the authors (one with a background in higher education career services and the other with training in environmental health), and were independently coded by two research assistants to improve inter-rater reliability. Themes that emerged are listed in Table 6. Example quotes are included in Appendix A.

Survey question: “What expertise or skills do you think will be needed to address the issue of climate change and human health in the next 10–20 years?” (open-ended responses, coded).

## 4. Discussion

The current state of the job market for public health graduates with training in climate change can be described as “emerging”. From the Indeed.com job description data analysis, we can see there are relatively few roles—even in search results from a broad-based job board with keywords focusing on public health and climate change—currently available for a graduate with a master’s level public health degree and a focus in climate change. Notwithstanding, it is likely that graduates would benefit from training in climate change-related competencies, even if the overt focus of their job is not directly related to climate change. Additionally, resonating with Wals, Corocoran, and others who frame educational institutions as change leaders, graduates with training in both climate change and public health can influence their institutions from within, to create systemic change in grappling with global warming.

The analysis of publichealthjobs.org data seemed to show that while jobs within public health that mention climate change or global warming were a very small proportion of the total, the fraction of such job postings had shown a statistically significant (*p* < 0.0001) increase over the last 16 years. This trend should be monitored by those involved in public health education and career placement of public health graduates, bearing in mind that while prior trends are often used to predict the future, they are not always the best indicator of future trends in a quickly changing world.

While “approaches for assessing, preventing and controlling environmental hazards that pose risks to human health and safety” [13] is not yet a core competency of public health degrees in the United States, the employer survey indicates that a large majority of respondents believe that there may be a growing need for graduates with training in climate change and health. The survey indicates that key skills include knowledge of climate mitigation, health equity and climate justice, an understanding of “downstream” effects of climate change, risk assessment, and technical skills in statistics, GIS mapping, and the carbon cycle. Comments from the responders indicate key themes focusing on these areas as well as communication (especially persuasive communication), finance/budgeting, cross-disciplinary collaboration and systems thinking, analytical skills, and an understanding of climate impacts on mental health, which resonate with Frankson et al.’s [53]. One Health Competency Domains including management, communication and informatics, values and ethics, leadership, team and collaboration, roles and responsibilities, and systems thinking. These skills also appear to be in alignment with the competencies proposed by ASPHER, CAPHIA, and the Global Consortium on Climate and Health Education.

Importantly, the scope and framing of this study focused primarily on the role of educational institutions in preparing graduates to solve the problems of today, and to meet the demands of today’s employers. Universities, however, not only provide education, produce research, and perform service to their communities; in addition, “higher education can play a pivotal role in turning society toward sustainability” [54]. This is an especially essential role in the face of massive and unpredictable global issues such as climate change. Universities create innovation, and can use their often privileged place in society to advocate for a sustainable future and to equip all of their graduates with understanding of their own environmental impact, both in the personal lives and in their careers. The challenges of climate change are profound enough to require an epistemological change; “sustainability is not just another issue to be added to an overcrowded curriculum, but a gateway to a different view of curriculum, of pedagogy, of organizational change, of policy and particularly of ethos” [54]. Additionally, following Scharmer’s Theory U, we note that knowledge itself is not in short supply; instead, there is a “knowing-doing gap: a disconnect between our collective consciousness and our collective action” and our entire “mental and social operating system” must be upgraded from “ego-awareness to eco-awareness” [55]. Therefore, while this article focused on historical trends and current and near-term workforce needs to attempt to predict, shape, and model the need for public health students with training in climate change, the disruptive reality created by climate change likely cannot be modeled through such methods. Education should therefore help graduates develop new capacities, allowing them to deal with disruptions and lead a transformational change. The issues of sustainability are so far-reaching that it can be argued that educational institutions must reframe their full mission, using sustainability as their foundation.

### Limtations

There are several limitations to the analysis. Indeed.com may not capture all jobs; some jobs are never posted; and the US-focused part of the site was the only section of the site analyzed. A re-examination of these findings over a longer period of time would be helpful. The publichealthjobs.org database has a self-selection bias towards employers specifically recruiting for public health, though this is part of the reason this database was selected for analysis; and the number of job postings mentioning climate change or global warming was sparse, but is useful in indicating trends over time. The employer survey was distributed to a convenience sample of employers, with certain industries/sectors overrepresented and with a likely bias towards those in the United States (especially those based near New York City). While the response rate of 14% appears to be low, it is comparable with other employer surveys in the public health field, where studies have included rates as low as 13.4% [20] and 19.5% [23]. It is important to note, as those in public health have observed from responses to crises such as Ebola and Zika outbreaks [56], funding—and thus the need to use this funding to quickly hire highly trained public health professionals—can change quickly, if and when current events or policy priorities shift. Thus, prior trends (such as a 16 year retrospective analysis of job postings) cannot be assumed to be an accurate indicator of future job market growth. Finally, there is a need for further research in this area; competencies required for tackling climate change also require students and employers to identify and adapt to uncertainty and change, and universities have a special role to play in creating transformative change and disruption using their own critical analyses.

## 5. Conclusions

Climate change is a growing threat to human health. While the current job market for candidates with training in both climate change and public health is relatively small, it appears to be growing; and it is likely that training in climate change competencies will increasingly benefit a range of public health organizations as climate change impacts continue to grow. Schools of public health can incorporate the skills and competencies related to climate change into their curricula and consider making them an integral/foundational part of the curriculum, if such training is not yet currently required. Employers, too, may benefit by taking note of the special intersection of skills and competencies offered by public health graduates with training in climate change-related issues. Graduates with such training can bring their paradigm-shifting lens to the work they do within any public health-related organization. Future research, including analyzing job postings, graduate employment outcomes, labor market projections, and employer surveys, could benefit curriculum development for educational institutions in countries around the world, and educational institutions could also remain at the forefront of the paradigm-shifting change that impacts the future public health workforce. By listening to the voices of current employers and assessing labor market trends, while also taking a wider view regarding the role of educational institutions in creating a sustainable world, these institutions can develop the skills and mindset needed to protect the public’s health from emerging challenges such as climate change.

## Figures and Tables

**Figure 1 ijerph-17-01310-f001:**
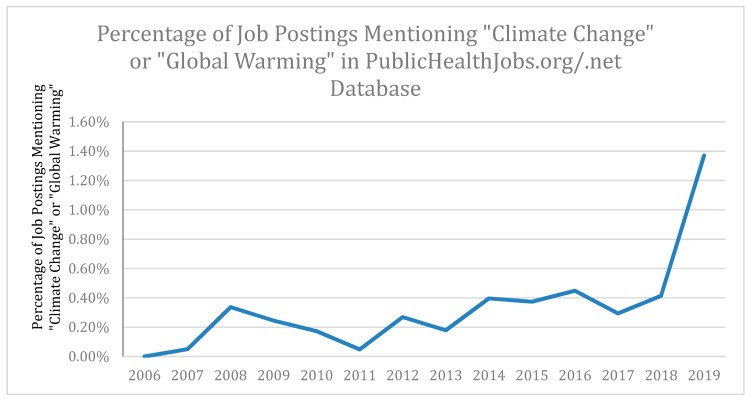
Percent of job postings mentioning “Climate Change” or “Global Warming” in the PublicHealthJobs.org database.

**Figure 2 ijerph-17-01310-f002:**
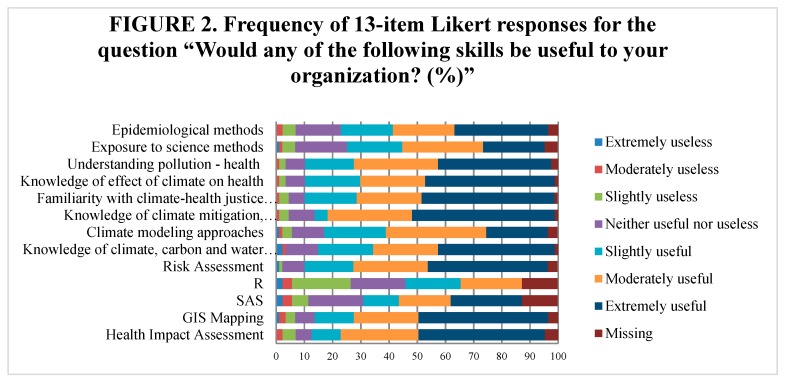
Frequency of 13-item Likert responses for the question “Would any of the following skills be useful to your organization?” (%). R and SAS refer to statistical analysis software.

**Figure 3 ijerph-17-01310-f003:**
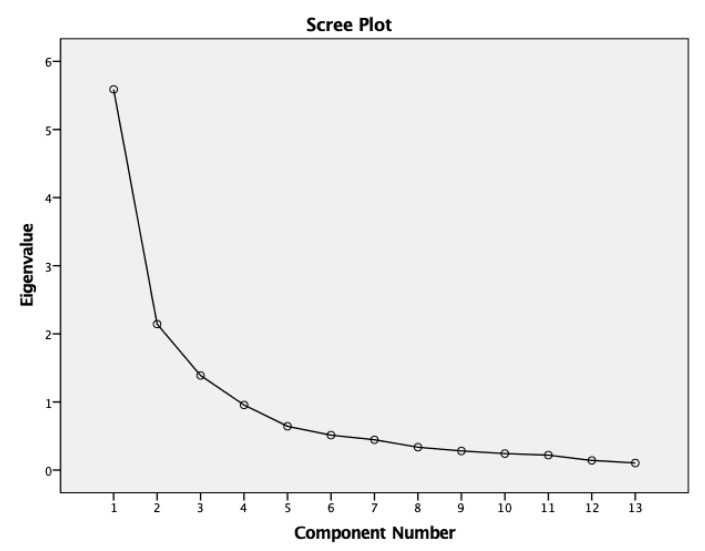
Scree plot of the final exploratory factor analysis (EFA) solution (three factors on 13 items).

**Table 1 ijerph-17-01310-t001:** Most common occupations in Indeed.com postings by the Standard Occupational Classification (SOC) code.

SOC Code	Occupation Title	Number
21-1099	Community and Social Service Specialists, All Other	36
25-1071	Health Specialties Teachers, Postsecondary	12
23-1011	Lawyers	11
19-2041	Environmental Scientists and Specialists, Including Health	9
27-3031	Public Relations Specialists	7
29-9012/29-9011	Occupational Health and Safety Technicians/Specialists	5
41-3099	Sales Representatives, Services, All Other	5
25-1053	Environmental Science Teachers, Postsecondary	5
19-3041	Sociologists	4
17-3029	Engineering Technicians, Except Drafters, All Other	4
11-1021	General and Operations Managers	4

**Table 2 ijerph-17-01310-t002:** Analysis of Data from Publichealthjobs.org/.net from 2003–2019.

Year	Total N Job Postings	Number of Jobs Mentioning “Climate Change”	Percentage
2003	116	0	0%
2004	899	0	0%
2005	998	0	0%
2006	1307	0	0%
2007	2006	1	0.05%
2008	2080	7	0.34%
2009	2044	5	0.24%
2010	2323	4	0.17%
2011	2095	1	0.05%
2012	2236	6	0.27%
2013	2236	4	0.18%
2014	2780	11	0.40%
2015	3213	12	0.37%
2016	2232	10	0.45%
2017	2041	6	0.29%
2018	2903	12	0.41%
2019	584	8	1.37%

**Table 3 ijerph-17-01310-t003:** Survey recipients vs. responders vs. Indeed.com postings.

Survey Recipients			Survey Responders		Indeed.Com Job Postings	
	Number	Percent	Number	Percent	Number	Percent
Corporation	194	36.13%	21	24.14%	32	20%
University	43	8.01%	9	10.34%	34	21.40%
Government	113	21.04%	32	36.78%	17	10.70%
Hospital	10	1.86%	1	1.15%	0	0
Nonprofit	172	32.03%	23	26.44%	76	47.80%
Unknown	5	0.93%	1	1.15%	0	0.00%

**Table 4 ijerph-17-01310-t004:** Pattern matrix of the EFA solution (three factors, 13 items).

Item		Factor		
		1	2	3
1	Health Impact Assessment	0.820		
2	GIS Mapping	0.607		
3	SAS			0.810
4	R			0.890
5	Risk Assessment	0.456		
6	Knowledge of climate, carbon and water cycles		0.928	
7	Familiarity with climate modeling approaches		0.885	
8	Knowledge of climate mitigation, adaptation and climate-health co-benefits		0.850	
9	Familiarity with climate-health justice issues	0.507	0.686	
10	Knowledge of direct, indirect and downstream effects of climate on health	0.655	0.501	
11	Understanding pollution-health consequences, causes and sources	0.791		
12	Exposure to science methods	0.772		
13	Epidemiological methods	0.645		
Rotation converged in 6 iterations.				
Extraction Method: Principal Component Analysis.				
Rotation Method: Varimax with Kaiser Normalization.				

**Table 5 ijerph-17-01310-t005:** Pattern matrix of the EFA solution (three factors, 13 items).

		Factor		
Item Would Any of the Following Skills be Useful to Your Organization? (%)		Population Health Exposure	Climate-Related Knowledge	Statistical Programming Language
1	Health Impact Assessment	0.820		
2	GIS Mapping	0.684		
3	SAS			0.829
4	R			0.914
5	Risk Assessment	0.534		
6	Knowledge of climate, carbon and water cycles		0.941	
7	Familiarity with climate modeling approaches		0.900	
8	Knowledge of climate mitigation, adaptation and climate-health co-benefits		0.838	
9	Understanding pollution-health consequences, causes and sources	0.776		
10	Exposure to science methods	0.799		
11	Epidemiological methods	0.666		
Cronbach’s α		0.832	0.897	0.761
Rotation converged in 5 iterations.				
Extraction Method: Principal Component Analysis.				

The weighted sum score is calculated by using the weighted variance percentage, ranges from −1.34 to 0.63.

**Table 6 ijerph-17-01310-t006:** Open-ended survey responses.

Skill	Number of Respondents Mentioning This Item
Communication/writing skills	19
Climate change knowledge	17
Public health expertise/training	17
Financing/Budgeting/Economic evaluation	13
Policy expertise/thinking	12
Critical thinking/Logical thinking/Systems thinking	12
Ecological/Agricultural/Geological/Environmental knowledge	11
Resilience and adaptation: Cross disciplinary understanding	9
Analytical skills	7
Marketing/Promoting/Advocacy	6

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
