# Peer review of "The Future of Careers at the Intersection of Climate Change and Public Health: What Can Job Postings and an Employer Survey Tell Us?"

_ijerph, 2020, doi:10.3390/ijerph17041310_

Round 1
Reviewer 1 Report
This is an interesting manuscript discussing a highly relevant topic. Overall, I feel that this manuscript suffers from too many different analyses of somewhat relevant data sets. I think the manuscript would be served by focusing on a couple of the data sets (the survey of employers and the publichealthjobs.org data) and eliminating the rest of the paper. The ASPPH data are generic to Environmental health graduates (compared to all MPH graduates) and the Bureau of Labor Statistics data are generic to public health jobs. They add nothing to the discussion of the need for competency in climate change. I'm not sure that I understand what the Indeed.com data add to the discussion.
In all, at best I would recommend focusing on the datasets mentioned above and provide a better description of the data collection methods and analyses. It is unfortunate that you ended up with a convenience sample of employers so that you can't calculate response rates. It limits the value of the study.
Reviewer 2 Report
I am not qualified to comment on the statistical robustness of the article so will confine my comments to other aspects of the paper. In this paper climate change competences are treated as an additional set of competences, which when added to the curricula may provide additional employment and employer benefits. Whist this may be so it totally ignores those (e.g. Wals, Concoran ) who argue that climate change education must be fundamentally disruptive, requiring a new paradigm of transformative social learning which focuses on addressing complexity and uncertainty (e.g. through systems and design thinking) and one in which the future is not an extrapolation of the past. Although this may be outside the scope of the paper this should at least be acknowledged. It could also usefully comment on the extent to which public health degree curricula do and should respond to employer’s needs (labour market demand) rather than attempt to drive change. The current conclusions section is somewhat brief and needs to be developed further – who needs to know this? what will they do with this information? what difference will it make to curricula and for addressing the impacts of climate change etc? The limitations section would also benefit from a consideration of the usefulness of a 16 year retrospective analysis and how this might indicate future trends.
Three additional minor points;
L270 – replace the word ‘crosswalk’ with another term more universally understood. Figure 2 quality needs to be improved to either show the actual data more clearly or redrawn to indicate the relative importance respondents attached to each answer. L375-377 Does the data actually indicate this or is it the authors wishful thinking?
Round 2
Reviewer 1 Report
Thank you for the changes you made to this manuscript. It now reads much better. My only recommendation would be to take the literature review out of the methods and results and include it in your introduction. It isn't a systematic review and would provide more insight and rationale why your study is needed. Otherwise, I have no other suggestions.
Minor edit, line 258 has a double period at the end of the sentence.
Author Response
Reviewer 1: We have moved the literature review from the methods section to the introduction (lines 60-68), and corrected the double period. Thank you!
Reviewer 2 Report
In response to my initial comments the authors have acknowledged those who feel that addressing the challenge of climate change in HE is about more than teaching new skills and competences but fundamentally about dealing with uncertainty. However, they qualify this immediately, noting the pragmatic imperative of also having to consider current employer needs, if students are to find employment. The remainder of the paper then continues on much as before.
Although the paper now notes the problems in universities associated with the perspective that the future can be predicted from past trends, this paper does just that. It also, on P376 notes that prior trends (such as a 16 year retrospective analysis of 376 job postings) are not always an accurate indicator of future job market growth, rather undermining its own concluding arguments. These additions have therefore made the paper incongruous.
To have sufficient academic merit for publication I feel the paper needs to address the above. It should reflect more deeply e.g. about the continued acceptance of a responsive demand led model of education in a situation where the challenge is fundamentally about dealing with uncertainty and where the future is not a progression of the past. They will need to also reinterpret the data, reflecting on its value as a historical data set.
Author Response
We have addressed the fact that the historical trends in the job board data may not be an accurate predictor of future trends, in line 349-351, noting that while the trend in the job postings over 16 years may be statistically significant, it may not predict the future due to the volatility caused by climate change. We also pointed out that past trends do not always predict the future specifically in the public health arena in lines 399-402.
We have incorporated the perspective regarding paradigm shifts to frame the article, including in lines 29-32 in the abstract, and added a paragraph to the discussion section focusing on the multiple roles of higher education institutions, including as change agents; and highlighting the paradigm shift which higher education institutions must be part of, referencing Corcoran and Wals as well as Scharmer’s Theory U, thus allowing this research to be framed in the context that it is based on an established paradigm which is now shifting, and noting that universities must prepare their graduates—and society—to adapt to unpredictable and disruptive change (lines 366-387 and line 406, 423-425).